# Acute Effects of Brevetoxin-3 Administered via Oral Gavage to Mice

**DOI:** 10.3390/md21120644

**Published:** 2023-12-16

**Authors:** Peggy Barbe, Jordi Molgó, Robert Thai, Apolline Urman, Denis Servent, Nathalie Arnich, Mathilde Keck

**Affiliations:** 1Université Paris Saclay, CEA, INRAE, Département Médicaments et Technologies pour la Santé (DMTS), SIMoS, 91191 Gif-sur-Yvette, France; peggy.barbe@cea.fr (P.B.); jordi.molgo@cea.fr (J.M.); robert.thai@cea.fr (R.T.); apolline.urman@cea.fr (A.U.); denis.servent@cea.fr (D.S.); 2Risk Assessment Directorate, ANSES—French Agency for Food, Environmental and Occupational Health and Safety, 94701 Maisons-Alfort, France; nathalie.arnich@anses.fr

**Keywords:** brevetoxin-3, oral gavage, mice, males, females, body temperature, symptomatology, ARfD

## Abstract

Brevetoxins (BTXs) constitute a family of lipid-soluble toxic cyclic polyethers mainly produced by *Karenia brevis,* which is the main vector for a foodborne syndrome known as neurotoxic shellfish poisoning (NSP) in humans. To prevent health risks associated with the consumption of contaminated shellfish in France, the French Agency for Food, Environmental and Occupational Health & Safety (ANSES) recommended assessing the effects of BTXs via an acute oral toxicity study in rodents. Here, we investigated the effect of a single oral administration in both male and female mice with several doses of BTX-3 (100 to 1,500 µg kg^−1^ bw) during a 48 h observation period in order to provide toxicity data to be used as a starting point for establishing an acute oral reference dose (ARfD). We monitored biological parameters and observed symptomatology, revealing different effects of this toxin depending on the sex. Females were more sensitive than males to the impact of BTX-3 at the lowest doses on weight loss. For both males and females, BTX-3 induced a rapid, transient and dose-dependent decrease in body temperature, and a transient dose-dependent reduced muscle activity. Males were more sensitive to BTX-3 than females with more frequent observations of failures in the grip test, convulsive jaw movements, and tremors. BTX-3’s impacts on symptomatology were rapid, appearing during the 2 h after administration, and were transient, disappearing 24 h after administration. The highest dose of BTX-3 administered in this study, 1,500 µg kg^−1^ bw, was more toxic to males, leading to the euthanasia of three out of five males only 4 h after administration. BTX-3 had no effect on water intake, and affected neither the plasma chemistry parameters nor the organs’ weight. We identified potential points of departure that could be used to establish an ARfD (decrease in body weight, body temperature, and muscle activity).

## 1. Introduction

In the Gulf of Mexico, particularly along Florida’s coast, blooms of the dinoflagellate *Karenia brevis* (*Dinophyceae*, formerly *Gymnodinium breve* and *Ptychodiscus brevis*) [1], also known as “red tides”, occur frequently, and regularly exert detrimental impacts on marine organisms, human health, and local economies [2,3,4]. These blooms are associated with the production of brevetoxins (BTXs, or PbTXs), and other potentially toxic metabolites [5,6,7].

BTXs constitute a family of heat-stable, lipid-soluble toxic cyclic polyethers, for the most part produced by *Karenia brevis* and other related *Karenia* spp. (*Karenia papilionacea*, *Karenia mikimotoi*, *Karenia longicanalis*, *Karenia icuneiformis,* and an undescribed species *Karenia* sp. 1), as well as by, among the raphidophytes, *Chattonella marina*, *Chattonella antiqua*, *Fibrocapsa japonica* and *Heterosigma akashiwo* [5,7,8,9].

Hort et al. [5] identified 88 compounds in the group of BTXs, most of them metabolites produced by marine organisms. BTXs accumulate in shellfish, fish and other marine organisms, and can be responsible for human poisoning in case of ingestion of food contaminated with BTXs. This syndrome is called “Neurotoxic Shellfish Poisoning” (NSP), and is characterized by neurological, gastrointestinal, and/or cardiovascular symptoms [4,7,10,11,12]. In addition, human toxicity can occur through the inhalation of BTXs-containing sea spray aerosols from red tide events [13,14,15], which cause airborne respiratory symptoms [16,17] characterized by cough, mucosal irritation and severe bronchoconstriction. Finally, human contamination was also reported through skin contact with a minor impact, with most studies showing only a very low percentage of the toxin penetrating through the skin [18,19]. In marine wildlife fauna, BTXs can cause immunotoxicity [20], and blooms of *Karenia* have been associated with fatality events of fish, manatees, sea lions, sea turtles, seabirds, but also terrestrial mammals (coyotes, ground squirrels, domestic dogs), and green tree frogs [21,22,23,24].

BTXs interact with voltage-gated sodium (Na_V_) channels and act as channel modulators in all cells and tissues in which the channels are expressed [25,26,27,28,29], albeit with different sensitivities [25,30]. BTXs are considered Na_V_ channel activators because (i) they shift activation to more negative membrane potential values, (ii) prolong the mean open time of unitary Na_V_ channels, (iii) inhibit channel inactivation, and (iv) induce channel sub-conductance states [31].

At present, BTXs are regulated outside Europe in the USA, Mexico, Australia and New Zealand [32,33,34,35,36]. However, BTXs are considered emerging biotoxins in European coasts [25], and are not regulated despite the proven health risk representing their presence in shellfish. This is mainly due to the lack of toxicological and epidemiological data required to set shellfish maximum levels. In French coasts, BTXs were detected for the first time in bivalve shellfish from a Mediterranean lagoon in Corsica Island [9], and by the EMERGTOX network (BTX-2, BTX-3) during the period of 2018–2022 [37].

Toxicological data from human cases of food poisoning or from laboratory animal experiments are too limited to derive an acute oral reference dose (ARfD) [38]. Based on a review of the literature on human NSP case reports after the consumption of BTX-contaminated shellfish, Arnich et al. [8] identified two acute Lowest Observed Adverse Effect Levels (LOAELs). Based on the work of Mc-Farren et al. [39], they recalculated a LOAEL in the range of 27–40.5 MU.person^−1^ (MU: mouse unit; they assumed a 10 g oyster flesh weight rather than 20 g, as assumed by McFarren et al. [39]). Based on the work of Hemmert [40], they identified a LOAEL of 0.3–0.4 MU.kg^−1^ bw. In BTX-3 equivalents (1 MU = 3.4 µg BTX-3 [41]), the acute LOAELs would be 92–138 µg BTX-3 eq.person^−1^ and 1.02–1.36 µg BTX-3 eq.kg^−1^ bw, respectively. Therefore, to prevent health risks associated with the consumption of contaminated shellfish in France, ANSES recommended a guidance level of 180 µg BTX-3 eq.kg^−1^ bw shellfish meat for the occurrence of BTXs in shellfish [8]. In addition, assessing the effects of BTXs via an acute oral toxicity study in rodents was recommended.

In order to provide toxicity data to be used as a starting point for establishing an ARfD, we decided to conduct an acute oral toxicity study in both male and female mice with BTX-3 (single administration, 48 h observation period) taking into account symptomatology and biological parameters such as the monitoring of body temperature. Our study was focused on BTX-3 because it is the main BTX analog found in shellfish (as reviewed in [8]). A preliminary account on a part of this work has been communicated at the 20th international conference on harmful algae, held in Hiroshima, Japan (5–10 November 2023) [42].

## 2. Results

### 2.1. General Observations

At the beginning of the procedure, male and female mice showed a high mean weight of 27.0 ± 0.3 g and 23.7 ± 0.3 g (Figure 1a), respectively. The mice were administered with different doses of BTX-3 (100, 250, 500, 750, 1,000 or 1,500 µg kg^−1^ bw) via oral gavage and observed for 48 h. We confirmed via LC-ESI-MS/MS that the concentration of the BTX-3 solutions administered to mice gave results in line with the expectations (Appendix B).

With the exception of three male mice administered with 1,500 µg kg^−1^ bw of BTX-3, all mice survived during the 48 h of the procedure. Three male mice administered with 1,500 µg kg^−1^ bw of BTX-3 had to be euthanized 4 h after the administration because they showed critical endpoints listed in the scoring table described in Section 4: (i) important decrease in body temperature, (ii) full-body stiffening, (iii) uninterrupted convulsive movements of the jaw, and (iv) repeated tremors. All mice administered with different doses of BTX-3 showed a normal water intake ≥ 1 mL 24 h^−1^ during the 48 h of the experiment (Appendix A).

### 2.2. Effect of BTX-3 on Quantitative Biological Parmeters

#### 2.2.1. Animal Weight

Male mice administered with 100 to 750 µg kg^−1^ bw of BTX-3 showed an increase in body weight over 48 h that was significantly different from the time before administration (i.e., T_0_) only for male mice administered with 100, and 500 µg kg^−1^ bw (Figure 1). Male mice administered with 1,000 µg kg^−1^ bw of BTX-3 showed a transient loss of weight after 24 h (−2.4% compared to T_0_) that was not observed after 48 h (Figure 1b). None of male mice showed a decrease in body weight ≥10%.

All female mice treated with BTX-3, whatever the dose, showed a decreased of body weight 24 h after administration (Figure 1b). This decrease in weight was more important and significantly different from T_0_ for female mice administered with 1,500 µg kg^−1^ bw of BTX-3 (Figure 1). Four females showed a decrease in weight ≥10% after BTX-3 administration: one administered with 250 µg kg^−1^ bw (−10.4% decrease in weight 48 h after administration), one administered with 750 µg kg^−1^ bw (−10.7% and −11.1% decrease in weight 24 h and 48 h after administration, respectively), two administered with 1,500 µg kg^−1^ bw (−10.5% and −11.5% decrease in weight 48 h after administration) (Figure 1b).

These results showed that the administration of the high dose of 1,500 µg kg^−1^ bw led to a significant loss of weight (−7.5%) in female mice 24 h after administration when compared to before administration. In addition, the administration of 1,000 µg kg^−1^ bw led to a loss of weight in male mice 24 h after administration when compared to before administration. Overall, these results indicated that 24 h might be a crucial time for BTX-3 to impact the animal weight.

#### 2.2.2. Body Temperature

All male and female mice administered with ≥250 µg kg^−1^ bw of BTX-3 showed a transient decrease in body temperature 2 h after administration that was proportional to the administered dose of BTX-3 (Figure 2).

Two hours after administration, the female mice administered with 100 µg kg^−1^ bw of BTX-3 showed a significant and transient decrease in body temperature of 0.8 ± 0.2 °C that was defined as a NOEL (no effect level) because the decrease was of less than 1 °C (Figure 2b). The opposite was observed for mice administered with 1,000 and 1,500 µg kg^−1^ bw of BTX-3. In fact, mice administered with 1,000 µg kg^−1^ bw of BTX-3 showed a significant decrease in body temperature, 1.9 °C in males and 2.3 °C in females, compared to the temperature at T_0_ (Figure 2b). Mice administered with 1,500 µg kg^−1^ bw of BTX-3 showed a huge decrease in body temperature of 4.4 °C in males and 3.4 °C in females; this decrease is significantly different from temperature at T_0_ in female mice (Figure 2b).

After 24 h and 48 h of administration, all mice administered with ≤1,000 µg kg^−1^ bw of BTX-3 maintained a normal temperature (Figure 2). Female mice administered with 1,500 µg kg^−1^ bw of BTX-3 still showed a significant decrease in body temperature, but defined as NOEL, of 1.0 ± 0.2 °C after 24 h, and 0.8 ± 0.1 °C after 48 h compared to temperature at T_0_.

#### 2.2.3. Clinical Chemistry and Organ Weight

At the end of the 48 h observation period, the mice were euthanized for plasma and organ collection in order to investigate the toxicological effect of BTX-3 on clinical chemistry and organ weight.

Hepatic, renal, pancreatic, metabolic, and muscle parameters, as well as electrolytes levels, were quantified. Regardless of the dose of BTX-3, no significant signs of perturbations of plasma chemistry were detected, with parameters remaining within the standard thresholds, suggesting the absence of systemic toxicity under these conditions (Appendix A). The only significant difference observed was between the glucose levels of male mice administered with 500 µg kg^−1^ bw of BTX-3 versus those of female mice administered with 1,000 µg kg^−1^ bw of BTX-3, which could probably be explained by randomization.

Similarly, BTX-3, whatever the dose, did not induce any modification in organ weight (Appendix A). The only significant differences observed were between male versus female kidney weight, which could be explained by both sex effect and randomization. Importantly, the three male mice administered with 1,500 µg kg^−1^ bw of BTX-3 and euthanized only 4 h after oral gavage for ethical reasons did not reveal any sign of asphyxia or organ damage during necropsy.

### 2.3. Effect of BTX-3 on Symptomatology

Mice administered with 100 and 250 µg kg^−1^ bw of BTX-3 showed no abnormal symptoms (Figure 3a–d).

One female mouse administered with 500 µg kg^−1^ bw of BTX-3 showed transient reduced muscle activity 24 h after administration; however, this effect was not observed 48 h after administration (Figure 3b).

One female mouse administered with 750 µg kg^−1^ bw of BTX-3 showed a transient failure in the grip test and reduced muscle activity after 2 h that disappeared after 24 h of administration (Figure 3a,b). Three male mice administered with 750 µg kg^−1^ bw of BTX-3 showed a reduced muscle activity after 2 h that was still maintained in one male after 24 h and disappeared after 48 h of administration (Figure 3b). Two of these three male mice also showed occasional jaw convulsive movements after 2 h that disappeared after 24 h of administration (Figure 3c).

Two male mice administered with 1,000 µg kg^−1^ bw of BTX-3 showed a partial failure in the grip test after 2 h that persisted after 24 h but disappeared after 48 h of administration (Figure 3a). Three male and all five female mice administered with 1,000 µg kg^−1^ bw of BTX-3 showed a reduced muscle activity after 2 h that persisted at 24 h for two males and then disappeared 48 h after administration (Figure 3b).

Three male mice administered with 1,500 µg kg^−1^ bw of BTX-3 rapidly failed in the grip test because of paralysis of the front (fore) and hind limbs (legs) (Figure 3a,b). They also showed jaw convulsive movements and repeated tremors (Figure 3c,d). For ethical reasons, these three male mice had to be euthanized 4 h after administration. The two other male mice administered with 1,500 µg kg^−1^ bw of BTX-3 showed a transient reduced muscle activity after 2 h that disappeared 24 h after administration (Figure 3b). All five female mice administered with 1,500 µg kg^−1^ bw of BTX-3 showed a reduced muscle activity after 2 h that persisted after 24 h for three females, and then disappeared 48 h after administration (Figure 3b). Only one female mouse administered with 1,500 µg kg^−1^ bw of BTX-3 showed transient convulsive jaw movements after 2 h that disappeared 24 h after administration (Figure 3c).

Overall, BTX-3’s impacts on symptomatology were rapid, appearing during the first 2 h after administration, and were transient, disappearing 24 h after administration. Male mice seemed to be more sensitive than female since failure in the grip test and the presence of jaw convulsive movements were more frequently observed in males (five males versus one female, and six males versus one female, respectively) (Figure 3a,c). In addition, BTX-3 administered at the high dose of 1,500 µg kg^−1^ bw induced repeated tremors but only in males (Figure 3d).

### 2.4. Identification of Potential Points of Departure for Establishing an ARfD

Table 1 summarizes the NOAELs (No Observed Adverse Effect Level) and LOAELs (Lowest Observed Adverse Effect Level) of the following critical endpoints: decrease in body weight, decrease in body temperature, alteration in muscle activity, grip test, jaw movements, and tremors. A benchmark dose approach was tested for some endpoints, but the criteria were fulfilled only for muscle activity.

The lowest point of departure would be a NO(A)EL of 100 µg kg^−^^1^ bw based on a decrease in body temperature and body weight in females. In a human equivalent dose (HED), this would correspond to 13.44 µg kg^−^^1^ bw with a mice body weight of 0.02284 kg (for this group of mice), and a default human body weight of 70 kg, according to the following formula:Human equivalent dose = Animal dose × (Animal weight/Human weight)^1/4^

To derive an ARfD, the following uncertainty factors would be applied:UFA (interspecies variability) = 2.5 (less than 10 because of the use of an HED)
UFH (inter-individuals variability) = 10

The derived ARfD would be 13.44/25 = **0.54 µg kg^−1^ bw** = **0.60 nmol kg^−1^ bw**. The corresponding maximum level in bivalves would be 94 µg kg^−1^ bivalve meat (more protective than the value of 180 µg kg^−1^ bw of bivalve meat proposed by Arnich et al. [8]).

Another potential point of departure would be an alteration in muscle activity, with a NOAEL of 250 µg kg^−1^ bw in females. In HED, the dose would be 33.92 µg kg^−1^ bw with a mice body weight of 0.02372 kg (for this group of mice) and a default human body weight of 70 kg. To derive an ARfD, the following uncertainty factors would be applied:UFA (interspecies variability) = 2.5
UFH (inter-individuals variability) = 10(1)

The derived ARfD would be 33.92/25 = **1.36 µg kg^−1^** = **1.52 nmol kg^−1^ bw**. The corresponding maximum level in bivalves would be 238 µg kg^−1^ bivalve meat (less protective than the value of 180 µg kg^−1^ of bivalve meat proposed by Arnich et al. [8]). For this endpoint, it was possible to calculate a BMDL10 for males and females. The lowest value was 435 µg kg^−1^ in males. In HED, the dose would be 60.42 µg kg^−1^ bw. The derived ARfD would be 60.42/25 = **2.42 µg kg^−1^ bw** = **2.70 nmol kg^−1^ bw**.

## 3. Discussion

We conducted an acute oral toxicity study in both male and female mice with increasing doses of BTX-3 for a 48 h observation period taking into account biological parameters and symptomatology, revealing the different effects of this toxin. It is worth noting that BTX-3 administered at the highest dose of 1,500 µg kg^−1^ bw was more toxic in males compare to females, leading to the euthanasia of three out of five males, just 4 h after administration. However, female mice seemed to be more sensitive to BTX-3’s impact on the loss of body weight since the administration of the toxin, whatever the dose and even at the lowest dose of 100 µg kg^−1^ bw, led to a loss of weight only in female mice 24 h after administration when compared to before administration. BTX-3 induced a rapid, transient and dose-dependent decrease in body temperature in both male and female mice.

In the literature, a study published in 1989 and conducted in male rats reported a dose-dependent decrease in body temperature following the intravenous administration of PbTX-2 (BTX-2) with doses from 12.5 to 100 µg kg^−1^ bw [43]. Interestingly, the temperature decrease occurred within the first 2 h following toxin administration, whatever the dose. These results indicate that rodents respond in a similar manner to BTXs despite differences in activity between BTX-2 and BTX-3 and distinct administration routes (intravenous in rats [43] and oral in mice in our study). In another study published in 2001, the acute intraperitoneal (i.p.) administration of BTX-3 at 180 µg kg^−1^ bw to female mice has been reported to cause hypothermia, and a delayed hyperthermic response detected using radiotelemetry monitoring of core temperature in undisturbed mice [44]. Other polyether marine toxins like ciguatoxins, which share a common receptor site with brevetoxins [45], produce an acute transient hypothermic response at sublethal doses in mice following i.p. or oral administration [46].

Regardless of BTX-3’s impacts on symptomatology, male mice seemed to be more sensitive than female mice since failure in the grip test and convulsive jaw movements were more frequently observed in males. Moreover, BTX-3 administered at the high dose of 1,500 µg kg^−1^ bw induced repeated tremors only in males. Noteworthy, BTX-3’s impacts on symptomatology were rapid, appearing during the first 2 h after administration, and were short-lived, disappearing 24 h after administration. Despite all these effects, BTX-3 neither influenced water intake of mice over the 48 h following administration, nor affected plasma parameters or apparent organ toxicity.

At the time this manuscript was submitted, an article dealing with the acute toxicology of BTX-3 in mice was accepted for publication, and became available as pre-proof [47]. Although BTX-3 was purchased from the same source, several differences in the study design were noted compared to the present study, which could impact the toxicological evaluation of the toxin. Concerning the animals, only inbred female mice were used in the published article, while our study included outbred males and females. It is interesting to note that in Costas et al.’s manuscript, the duration of fasting was 15 h, but the mice had access to glucose serum, whereas in the present study, fasting was limited to 3 h in accordance with OECD guidance 425 on acute oral toxicity. Furthermore, in the published paper, ethanol was used to dilute BTX-3 (at an unspecified concentration) and the volume of oral gavage was not defined, whereas in our study, BTX-3 was diluted in ≤5% DMSO and the toxin was administered at 10 µL g^−1^. Finally, other differences included the use of some subjective parameters (such as ataxia, lordosis, abnormal behavior) to assess the BTX-3 effect, whereas our study included quantifiable scores detailed in Table 2. The results of this recent publication did not in any way reveal morphological alterations, inflammation, necrotic areas, abnormal sizes or loss of normal structure. As in our study, they found no effect on blood parameters.

As far as we know, no acute oral reference dose (ARfD) has been established for BTXs so far due to the limited quantitative data both in experimental animals and related to human intoxications. The present study was specifically designed to provide data that could be used to establish this type of health-based guidance value. Based on the most sensitive endpoints from our results, we identified several potential points of departure (decrease in body weight, body temperature, muscle activity) and we proposed different options with the corresponding uncertainty factors. Our results could now be considered by a public health agency in order to establish an official ARfD.

Based on an ARfD, it is possible to estimate the maximum contamination level of BTX-3 in shellfish in order to protect public health. In the Codex Alimentarius standard for live and raw bivalve mollusks (CODEXSTAN 292–2008, rev. 2015) [48], the maximum level for BTXs is 200 mouse units (MU) or equivalent per kg of mollusk flesh. An MU is the amount of raw extract required to kill 50% of mice using a mouse bioassay [49]). The United States, Australia, New Zealand, and Mexico apply this threshold, converted in BTX-2 to 800 µg kg^−1^ shellfish flesh [32,33,34,35,36]. In France, ANSES concluded that the Codex maximum level does not appear protective enough and recommended a guidance level of 180 µg BTX-3 eq kg^−1^ bw shellfish meat [8]. The French maximum level was derived from two acute Lowest Observed Adverse Effect Levels (LOAELs), based on a review of the literature on human NSP case reports after consumption of BTX-contaminated shellfish. The calculation included a default protective consumption level of 400 g of shellfish flesh per person as set by the EFSA to protect the largest shellfish consumers [50]. This portion was in accordance with consumption levels of French high consumers of seafood products. Based on the ARfD that could be derived from the potential points of departure identified in the present study, it would be possible to estimate the maximum level of BTX-3 in shellfish in order to protect public health, and to compare this value with existing thresholds at the national or international level.

Further analyses of the toxicological effects of BTX-3 would be interesting to perform to gain a deeper understanding of the time-lapsed symptomatology. Thus, it would be interesting to use video monitoring in order to avoid the animal manipulation effect, and to be able to review observations. It would also be relevant to identify the precise time of symptoms, which will provide important information in terms of toxicokinetics. Moreover, it could be informative to explore the cardiovascular impacts of BTX-3 administration to rodents.

Such acute oral toxicity data would also be needed for other major BTX analogs found in shellfish, and would allow researchers to determine relative potency factors compared to the most toxic analog of the BTX family. In addition, long-term effects following repeated oral exposure should be investigated in order to propose a chronic health-based guidance value.

## 4. Materials and Methods

### 4.1. Chemicals

PbTx-3/BTX-3 (product code: L8902, batch number: 901.122L8902, purity: ≥95%, and MW: 897.11 g mol^−1^) was provided by Latoxan (Portes-lès-Valence, France). The product was stored at −20 °C according to the supplier’s indications. The solutions for administration were freshly prepared and used in glass bottles with ≤5% DMSO for successive dilutions, as described below:2 aliquots, each containing 1 mg of BTX-3 in powder, were received in glass bottles and stored at −20 °C.333 μL of 100% DMSO were added to each aliquot.The total volume of each aliquot was pooled to constitute a “stock solution of BTX-3 at 3 mg mL^−1^ in 100% DMSO”. −For solution A at 150 ng μL^−1^: 600 μL of the stock solution of BTX-3 at 3 mg mL^−1^ was diluted with 11.4 mL of NaCl 0.9% into a glass bottle. 10 μL g^−1^ of this solution was administered via oral gavage, meaning 1,500 μg kg^−1^.−For solution B at 100 ng μL^−1^: 3 mL of the solution A was diluted with 1.5 mL of NaCl 0.9% into a glass bottle. 10 μL g^−1^ of this solution was administered via oral gavage, meaning 1,000 μg.kg^−1^.−For solution C at 75 ng μL^−1^: 5 mL of the solution A was diluted with 5 mL of NaCl 0.9% into a glass bottle. 10 μL g^−1^ of this solution was administered via oral gavage, meaning 750 μg kg^−1^.−For solution D at 50 ng μL^−1^: 5 mL of solution C was diluted with 2.5 mL of NaCl 0.9% into a glass bottle. 10 μL g^−1^ of this solution was administered via oral gavage, meaning 500 μg kg^−1^.−For solution E at 25 ng μL^−1^: 3 mL of solution D was diluted with 3 mL of NaCl 0.9% into a glass bottle. 10 μL g^−1^ of this solution was administered via oral gavage, meaning 250 μg kg^−1^.−For solution F at 10 ng μL^−1^: 2 mL of solution E was diluted with 3 mL of NaCl 0.9% into a glass bottle. 10 μL g^−1^ of this solution was administered via oral gavage, meaning 100 μg kg^−1^.

### 4.2. LC-ESI-MS/MS Checking of Items Used for Administration to Mice

Standard solutions (PbTx-3, code L8902, batch 901.122 for calibration curve, and PbTx-2, code B-6850, batch 0251-1 as internal MS standard) were prepared at an initial concentration of 1 mM and 100 µM in DMSO, respectively. All HPLC solvents and buffer reagents were of analytical grade. To establish the calibration curve, known amounts of PbTx-3 were spiked in the range of concentrations from 10 to 200 µM into 50% MeOH/water containing 2 mM ammonium formate and 50 mM formic acid as buffer, and a fixed amount of PbTx-2 as the internal standard (IS). The same amount of IS was also spiked into unknown solutions to be dosed.

The dosages of PbTx-3 versus PbTx-2 as internal standard were performed via LC-ESI-MS/MS. The LC-MS system consisted of an Agilent 1100 HPLC coupled online to an Esquire-HCT ion trap mass spectrometer (Bruker-Daltonics, Billerica, MA, USA) equipped with an electrospray ionization (ESI) source. LC separation was carried out at a flow rate of 300 µL/min with a Phenomenex reverse phase Kinetex 5 µm C18 100Å column (2.1 × 100 mm) with its SecurityGuard precolumn maintained at 40 °C. Mobile phases consisted of water (A), and methanol/water (95: 5, *v*/*v*) (B), both containing 2 mM ammonium formate and 50 mM formic acid. The gradient used was 30–80% B over 1 min, 80–95% B over 7 min, 95–100% B for 1 min and maintained for 1 min, and then 100–30% B over 0.1 min, held for 3 min for equilibration. The injection volume was 10 µL.

The ESI conditions were optimized for the detection and fragmentation of PbTx-3 and PbTx-2 in high-resolution mode (Standard Enhanced). Data analysis software (Bruker Daltonics) was used for the qualitative and quantitative processing of the raw MS and MS/MS data. The ion chromatograms of the m/z of both toxins (*m/z* 897.5 for PbTx-3 and *m/z* 895.5 for PbTx-2) were extracted as EICs (Extracted Ion Chromatograms). The EIC peaks were integrated and the ratio of PbTx-3/PbTx-2 enabled us to generate PbTx-3 calibration curves and to further determine the concentration of PbTx-3 in unknown samples. The fragmentation profile of each *m/z* was used to confirm the identity of each toxin, as well as the LC elution retention time.

### 4.3. Authorizations and Animals for In Vivo Experiments

The procedure was validated by the local animal ethics committee (CETEA DSV—comité n°44) and received the authorization APAFIS#32015-2021061612169288 v1. The experiments were carried out using male 4-week-old and female outbred Swiss mice (Janvier Laboratory, St. Berthevin, France), which are a practical and commonly used model for pre-clinical toxicity studies. The animals were housed in polysulfone cages under standard conditions: room temperature (20 ± 2 °C), hygrometry (55 ± 10%), light/dark cycle (12 h/12 h), air replacement (15–20 volumes.h^−1^), and drinking water and pelleted maintenance diet (LASQCdiet^®^ Rod16, Genobios, France) ad libitum. During the acclimation period, the animals were housed in groups of 5 of the same sex, whereas from T_0_ and during the 48 h of the experiment, the animals were housed individually to follow drinking consumption.

### 4.4. Procedure for In Vivo Experiments

The mice (*n* = 30 males, and *n* = 30 females) were scored and randomized before administration via oral gavage with 10 µL g^−1^ (199–312 µL per mouse) of BTX-3 at 6 different doses (100, 250, 500, 750, 1,000, or 1,500 µg kg^−1^ bw) with sterile plastic feeding tubes, 20 ga × 38 mm (Phymep, Paris, France) (Figure 4). The doses and time points used in this study were chosen according to a pilot study (described in Appendix C) that was conducted in compliance with 3Rs rule with a low number of female Swiss mice (*n* = 7) and four distinct doses of BTX-3 (500, 750, 1,000, or 2,000 μg kg^−1^ bw). The mice were carefully observed following the 4 h after administration, and then at 24 h, and 48 h after administration. At 2 h, 24 h, and 48 h, a graded score was applied for the following critical endpoints listed in Table 2: decrease in body weight, decrease in body temperature, decrease in water intake, alteration in muscle activity, failure to grip test, jaw movements, and tremors. All parameters were stated in vigil animals. The grip test was performed with equipment used to place a mouse on a grid (Appendix A), and the grid was then turned over (Appendix A) to observe the success or failure with the attempted collision, or failure without the attempted collision under the 5 s after the grid was turned around.

**Table 2 marinedrugs-21-00644-t002:** Scoring table.

Observed Parameter	Grade 1 *	Grade 2 *	Grade 3 *	Grade 4 *
** Weight relative to T_0_ **	<−10%	−10% to −20%	−20% to −30%	>−30%
** Temperature (anal, vigil) **	Normal	↘ during 24 h	↘ during 48 h	N.A.
** Water intake **	≥1 mL 24 h^−1^	<1 mL 24 h^−1^	<2 mL 48 h^−1^	N.A.
** Muscle activity **	Normal	Reduced/mouse in sitting position	Reduced/crossed front legs/paralysis of hind legs	Full body stiffening
** Grip test **	Success	Failure with attempted collision	Failure without attempted collision	N.A.
** Jaw movements **	Normal	Occasional convulsive movements	Repeated convulsive movements	Uninterrupted convulsive movements
** Tremors **	No	Occasional	Repeated, but not intense	Repeated and intense

* **Grade 1** was considered normal. **Grade 2** meant that mouse must be monitored. **Grade 3** meant that mouse must be monitored more closely with addition of semi-liquid food and heating on mat. If four or more grade 3 parameters were observed, mice were euthanized. **Grade 4** meant that mouse must be euthanized. N.A.: not applicable. ↘ decrease.

### 4.5. Samples and Analysis

At 48 h after administration, the mice were anesthetized for blood collection and plasma analysis. The collected plasma samples were freshly analyzed on a Piccolo Xpress^®^ (Abaxis Europe GmbH, Griesheim, Germany) with AmLyte 13 disk to determine the following parameters: glucose, blood urea nitrogen, bilirubin, albumin, alanine aminotransferase, aspartate aminotransferase, creatine kinase, sodium, potassium, and calcium. Then, the mice were euthanized via cervical dislocation and dissected for macroscopic observation of their organs. Brain, heart, lungs, thymus, liver, spleen, intestines, kidneys, and urinary bladder were observed to identify signs of toxicity. After macroscopic observation, the brain, heart, lungs, small intestine, large intestine (including caecum), right kidney, and left kidney were weighted in the distinct group of mice.

### 4.6. Data Processing and Statistical Analysis

Quantitative data are shown as means, with error bars indicating the standard error of the mean (SEM). Normality was assessed using the d’Agostino–Pearson test. Comparisons between more than two groups were performed using the Kruskal–Wallis non-parametric test followed by Dunn’s multiple comparisons test. To compare repeated measurements over time, we used the non-parametric mixed-effects model followed by Dunnett’s multiple comparisons test. Differences were considered significant if the *p* value was <0.05. Statistical analyses and figures were performed using GraphPad Prism 9 software (GraphPad software Inc., San Diego, CA, USA).

Benchmark dose modeling was conducted with a web application of PROAST version 70.1 (copyright RIVM National Institute for Public Health and the Environment, https://www.rivm.nl/en/proast, accessed on 7 August 2023), a software package for the statistical analysis of dose–response data. Its main purpose is the dose–response modeling of toxicological data, and the derivation of a benchmark dose (BMDL) in human risk assessment.

## Figures and Tables

**Figure 1 marinedrugs-21-00644-f001:**
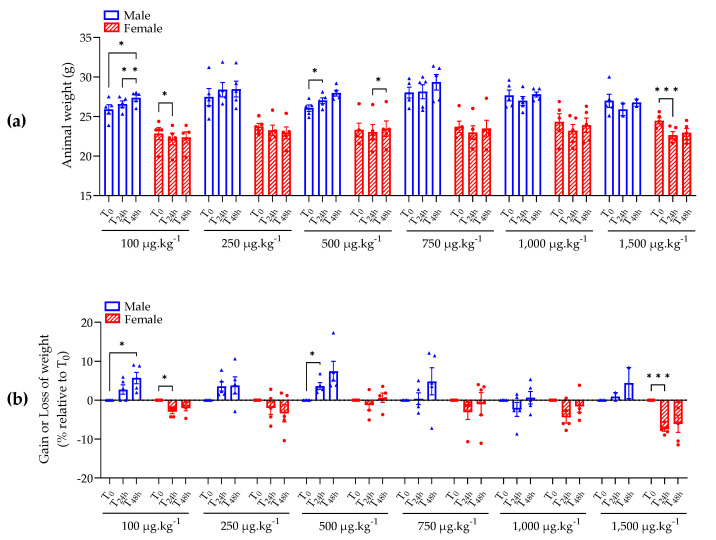
Swiss mouse body weight (**a**) and gain or loss of weight (**b**) after BTX-3 oral administration for each group (BTX doses are indicated, female are represented in red striped histograms) over 48 h. Mixed-effects model followed by Dunnett’s multiple comparisons test was used; * *p* < 0.05, ** *p* < 0.01, *** *p* < 0.001; *n* = 5 mice/group, except for male at 1,500 µg kg^−1^ bw, where three mice were euthanized at T_4h_.

**Figure 2 marinedrugs-21-00644-f002:**
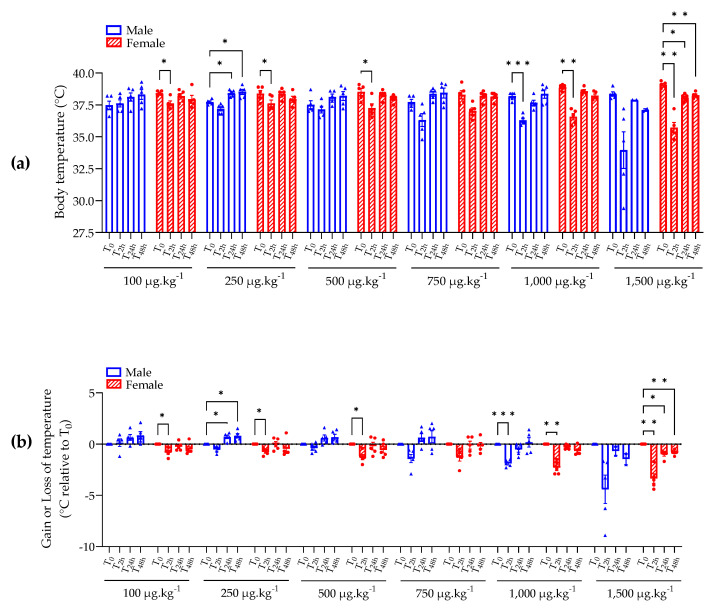
Swiss mouse temperature (**a**) and gain or loss of temperature (**b**) after BTX-3 oral administration for each group (BTX doses are indicated, female are represented in red striped histograms) over 48 h. Mixed-effects model followed by Dunnett’s multiple comparisons test was used; * *p* < 0.05, ** *p* < 0.01, *** *p* < 0.001; *n* = 5 mice/group, except for male at 1,500 µg kg^−1^ bw, where three mice were euthanized at T_4h_.

**Figure 3 marinedrugs-21-00644-f003:**
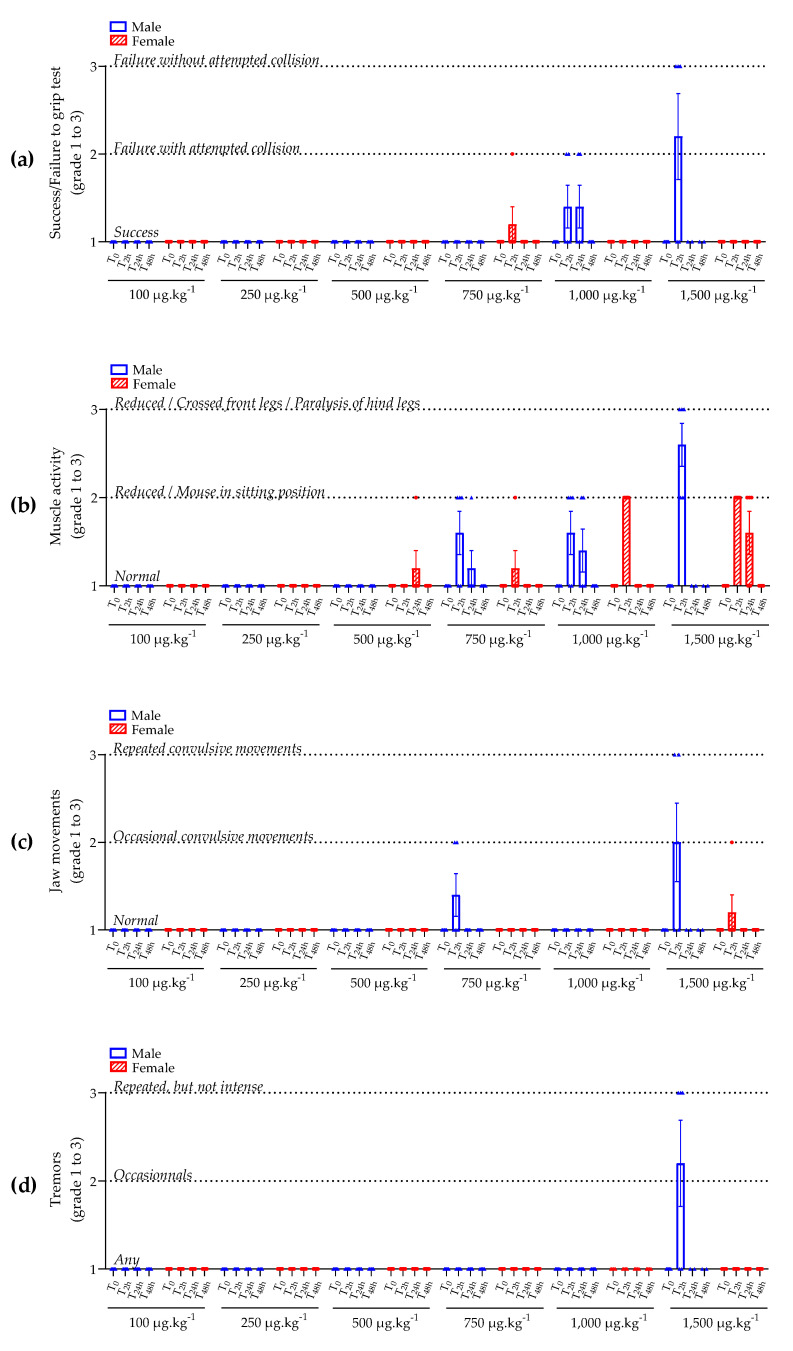
Swiss mouse success or failure in grip test (**a**), muscle activity (**b**), jaw movements indicating difficulty to breath (**c**), and tremors (**d**) observed after BTX-3 oral administration for each group (BTX doses are indicated, female are represented in red striped histograms) over 48 h. A score was attributed to symptomatology, with grade 1 corresponding to normal symptomatology, grade 2 to moderate modifications in symptomatology, and grade 3 to important modifications in symptomatology. No statistical tests could be applied; *n* = 5 mice/group, except for male at 1,500 µg kg^−1^ bw, where three mice were euthanized at T_4h_.

**Figure 4 marinedrugs-21-00644-f004:**
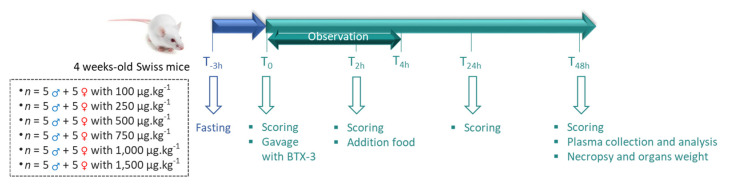
Schematic representation of the experimental procedure consisting of acute oral gavage administration with purified BTX-3 to male and female mice, followed by scoring at the three indicated time periods.

**Table 1 marinedrugs-21-00644-t001:** NOAELs and LOAELs identified from this study for critical endpoints.

Endpoint	NOAEL	LOAEL	BMDL	Comment
** Decrease in body weight **	100 µg kg^−1^ in ♀	250 µg kg^−1^ in ♀	Not possible *The 500* µg kg^−1^ *group does not respond sufficiently and distorts the dose–response relationship*	Adverse effect: 1 ♀ at −10.4%.
** Decrease in body temperature **	NOEL: 100 µg kg^−1^ in ♀	LOEL: 250 µg kg^−1^ in ♀	Not possible	Decrease of 0.8 ± 0.2 °C at 100 µg kg^−1^ defined as a NOEL (no effect level) because the decrease is of less than 1 °C.
** Muscle activity **	250 µg kg^−1^ in ♀ 500 µg kg^−1^ in ♂	500 µg kg^−1^ in ♀ (1 ♀ had a score of 2 at T_24h_). 750 µg kg^−1^ in ♂	BMDL10♀ = 448 µg kg^−1^ BMDL10♂ = 435 µg kg^−1^	BMDL10 for incidence of animals with a score >1.
** Grip test **	500 µg kg^−1^ in ♀	750 µg kg^−1^ in ♀	BMDL was not calculated because these endpoints were less sensitive compared to body weight, body temperature and muscle activity.	
** Jaw movements **	500 µg kg^−1^ in ♂	750 µg kg^−1^ in ♂		
** Tremors **	1,000 µg kg^−1^ in ♂	1500 µg kg^−1^ in ♂		

NO(A)EL: No Observed (Adverse) Effect Level; LO(A)EL: Lowest Observed (Adverse) Effect Level; BMDL: benchmark dose lower confidence limit.

## Data Availability

Data supporting reported results can be found in the Appendix A.

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
