# Peer review of "Acute Effects of Brevetoxin-3 Administered via Oral Gavage to Mice"

_marinedrugs, 2023, doi:10.3390/md21120644_

Round 1

Reviewer 1 Report

Comments and Suggestions for Authors

This manuscript describes some of the effects of exposure to brevetoxins by oral gavage on mice with the aim of advising regulatory limits for emerging toxins in France. While not particularly novel, the manuscript provides valuable information to assist in risk assessments. 

line 59-the authors list routes of exposure to brevetoxins and include dermal exposure on this list. Dermal exposure is pretty minor and most studies show very low % of dose penetrating the skin. Listing this route along with others may give the reader the false impression that dermal exposure is as important as oral or inhalation. It is most decidedly not. I think this statement should be qualified by putting into context with oral or inhalation exposure. 

The authors state that this is the first study of Btx-3 administered by acute oral gavage to male and female mice, monitoring symptomology. This may be true. Most of the studies that this reviewer has read are on female mice only,  but there are some other similar studies which should be cited for comparison. These include:

Gordon, C.J., Kimm-Brinson, K.L., Padnos, B. and Ramsdell, J.S., 2001. Acute and delayed thermoregulatory response of mice exposed to brevetoxin. Toxicon39(9), pp.1367-1374.

Costas, C., Louzao, M.C., Raposo-García, S., Vale, C., Graña, A., Carrera, C., Cifuentes, J.M., Vilariño, N., Vieytes, M.R. and Botana, L.M., 2023. Acute toxicology report of the emerging marine biotoxin Brevetoxin 3 in mice: Food safety implications. Food and Chemical Toxicology, p.114178.

Finally, it would be helpful to put these results into context with the regulatory limits from other countries which currently regulate these toxins. 

I am suggesting some other very minor revisions 

Line 28 "3 on 5 males". Do the authors mean 3 out of 5 males?

Line 47 change "associated to" to "associated with"

Author Response

We thank all the reviewers for their insightful assessment of our manuscript. All of the valuable reviewers’ comments have been taken into account to improve the quality of our initial paper.

To facilitate the identification of the changes made within the manuscript, the modifications have been highlighted in yellow. Below are listed the changes in the manuscript, as well as our answers to the concerns raised by reviewers to facilitate your final decision.

We hope that such modifications might render our revised manuscript suitable for publication in Marine Drugs.

Yours sincerely, 

Mathilde Keck

Reviewer 1

This manuscript describes some of the effects of exposure to brevetoxins by oral gavage on mice with the aim of advising regulatory limits for emerging toxins in France. While not particularly novel, the manuscript provides valuable information to assist in risk assessments. 

Line 59-the authors list routes of exposure to brevetoxins and include dermal exposure on this list. Dermal exposure is pretty minor and most studies show very low % of dose penetrating the skin. Listing this route along with others may give the reader the false impression that dermal exposure is as important as oral or inhalation. It is most decidedly not. I think this statement should be qualified by putting into context with oral or inhalation exposure. 

Answer: We thank the reviewer for this relevant recommendation that we followed by dissociating “dermal exposure” from “oral or inhalation exposure”. We added a new sentence with novel references (18-19) describing the minor impact of BTXs’ contamination through skin contact in page 2, lines 55-57.

  1. Kemppainen, B.W.; Mehta, M.; Stafford, R.; Riley, R.T. Effect of Vehicle on Skin Penetration and Retention of a Lipophilic Red Tide Toxin (PbTx-3). Toxicon 1992, 30, 931–935, doi:10.1016/0041-0101(92)90393-j.
  2. Mehta, M.; Kemppainen, B.W.; Stafford, R.G. In Vitro Penetration of Tritium-Labelled Water (THO) and [3H]PbTx-3 (a Red Tide Toxin) through Monkey Buccal Mucosa and Skin. Toxicol Lett 1991, 55, 185–194, doi:10.1016/0378-4274(91)90133-q.

The authors state that this is the first study of Btx-3 administered by acute oral gavage to male and female mice, monitoring symptomology. This may be true. Most of the studies that this reviewer has read are on female mice only, but there are some other similar studies which should be cited for comparison. These include:

Gordon, C.J., Kimm-Brinson, K.L., Padnos, B. and Ramsdell, J.S., 2001. Acute and delayed thermoregulatory response of mice exposed to brevetoxin. Toxicon39(9), pp.1367-1374.

Costas, C., Louzao, M.C., Raposo-García, S., Vale, C., Graña, A., Carrera, C., Cifuentes, J.M., Vilariño, N., Vieytes, M.R. and Botana, L.M., 2023. Acute toxicology report of the emerging marine biotoxin Brevetoxin 3 in mice: Food safety implications. Food and Chemical Toxicology, p.114178.

Answer: We would like to thank the reviewer for this important comment. We now implemented the discussion with the two cited references (pages 9-10, lines 276-279 and 291-307). We described that a similar decrease in body temperature was also observed in several studies (new references 43, 44, 45, and 46) after intravenous or intraperitoneal or oral administration of BTX-2 or BTX-3 or ciguatoxins in mice or rat. We also compared our study protocol to the recent study from Costas et al. (new reference 47) carried out in inbred female mice with BTX-3 diluted in ethanol after 15 h of fasting duration and including not quantifiable parameters. 

Finally, it would be helpful to put these results into context with the regulatory limits from other countries which currently regulate these toxins. 

Answer: We thank the reviewer for this suggestion to put the results of this study into context with the regulatory limits from other countries that currently regulate these toxins. We discussed this point based on three novel references (48 to 50) included in the discussion in page 10, lines 316-333.

I am suggesting some other very minor revisions 

Line 28 "3 on 5 males". Do the authors mean 3 out of 5 males?

Answer: We do mean “3 out of 5 males”, therefore we replaced “3 on 5 males” by “3 out of 5 males” (page 1, line 28 and page 9, line 264).

Line 47 change "associated to" to "associated with"

Answer: We changed "associated to" to "associated with" (page 1, line 39).

Reviewer 2 Report

Comments and Suggestions for Authors

The ARfD of a toxin is an estimate of the amount of substance in food that can be ingested in a period of 24 h or less without appreciable health risk to the consumer based on all known facts at the time of the evaluation.

If the goal of this study is the identification of potential points of departure to establish an acute oral reference dose (ARfD) why the experiment last 48h? Author should explain.

Also, most of the BTX3 effects were observed in the first 2 h but at 24 h decreased and at 48 were even not detected. The experimental time seems too long. In any case, it would be interesting to characterize clinical signs, biochemical or histopathological effects over the time course following acute exposure. 

-The study lacks control animals, which doesn’t allow to stablish the statistically significant toxin effect or to derive the appropriate points of departure for characterizing safe doses in humans. In this way, it is difficult to know if the lowest doses had an effect when compared to non-treated mice. For instance, mice body temperature not only varies throughout the day, but could also vary from group housing versus single housing animals. Therefore, it is suggested to perform an assay in control mice to assess whether the environmental conditions could be affecting some of the measured parameters.

Besides, in some issues authors seems to be more focused in differences between male and female mice than in the BTX3 effect.

Also, it is unclear why authors selected 2h, 24 and 48 h for mice observation, must clarify the experimental reason for that. To stablish the NOAEL or LOAEL of the critical end points it would be beneficial to monitor effects in different timepoints between 2h and 24 h.

Finally, there is no discussion, just a description of results. More interpretation and discussion of the findings are needed.

Specific comments:

Page 2.

Line 73

- “emergent” should be changed for “emerging”.

Line 92-93

-  Authors should explain why a single administration and 48 h observation period when ARfD is stablished in a period of 24 h

Page 3.

Line 103

-Since water intake is presented, was food intake also measured?

Besides, It could be interesting in the experiment to quantify urine and feces.

Line 104-106.

- It is stated that three male mice (1,500 μg.kg-1 bw) had lower water intake and they survived for 48 h, whereas in the following sentence it is stated that another three male mice had to be euthanized. In contrast, in Figure 4 it is indicated that this dose was administered to “n = 5 males”. This should be explained.

Line 106-110

-Three male mice administered with 1,500 μg.kg-1 bw of BTX-3 had to be euthanized 4 h after the administration. Did the authors euthanize any male in the pilot study?

Line 115-117

How authors interpretate the transient low of weight after 24 h of male that received 1,00 μg.kg-1 bw of BTX-? May be 24 h is a crucial time that should study.

Line 128-131.

Authors could not stablish that female mice seemed to be more sensitive to BTX-3 impact on loss of weight since the administration of the high dose of 1,500 μg.kg-1 bw of BTX-3 led to a significant loss of weight. Remember, that 3 of 5 male mice treated with 1,500 μg.kg-1 bw of BTX-3 were euthanized at 4 h. 

Page 4

Line 142-144

-The significant differences in female body weight and body temperature decrease for 100 μg.kg-1 bw indicate that BTX-3 may still has an effect. Therefore, lower doses should be tested to assess at which dose the toxin does not cause adverse effects.

Page 5

Line 170-172

Since Authors found a significant difference between the glucose levels of male mice administered with 500 µg/kg and female mice administered with 1000 µg/kgof BTX3 it would be beneficial to characterize food intake to contextualize the findings.

Page 6 Line 182-184

Female administered with 500 µg/kg of BTX3 showed transient reduced muscle activity 24 h after administration, and this effect was not observed 48 h after administration.

It could be interesting to evaluate necropsy at 24h.

Page 6 208-213

Related to abnormal symptoms. Authors indicated that BTX3 impacts on symptomatology were rapid, appearing during the first 2 h. Did the authors observe the mice during the first 2h? As symptoms disappear 24 h after administration, did the authors find any abnormal symptoms between 2 and 24 h?

If 3 males were euthanized at 4h. What where the symptoms of the other mice at that time?

Page 8

-In Table 1 it is indicated 100 μg.kg-1 bw as the NOAEL for body weight and body temperature, whereas significant differences at 24 and 2 h respectively in Figures 1 and 2. Therefore, this dose should be not suggested as a NOAEL. This should be modified.

before defining 100 μg.kg-1 bw as NOAEL.

Line 232-233

Authors should clarify the points of departure for stablishing ARfD. For instance, about muscle activity authors should explain why is 500 μg.kg-1 bw in males.

Page 9

Discussion.

There is no discussion, just a description of results with some contradictions. It is not clear if BTX3 is more toxic to male than female mice.

Line 284-286

As authors pointed out that it would be relevant to identify the precise time of symptoms. Therefore, a study with more time points of test and symptoms evaluations must be performed.

Page 10

Materials and Methods

-Line 299, more details on toxin preparation and administration should be provided.

Line 336-338.

-It should be explained at which timepoint animals were caged individually.

Page 11

Line 354-357

Why the grip test was performed under only 5 sec after the grid was turned around? Please explain. More

-In the Section 4.4. Procedure for in vivo experiments, it is commented that a preliminary study was conducted to select the doses. It would be interesting that those results were also added to the manuscript.

Supplementary Material

- Figure S3 significant differences in kidneys weight between treatments is shown. How could be this related to brevetoxins? It would be interesting to evaluate histological slices to determine whether these changes translate into organ damage. Moreover, organ weight of control animals would be helpful to address whether these changes are related to the toxin or are the result of sex dimorphism.

Author Response

We thank all the reviewers for their insightful assessment of our manuscript. All of the valuable reviewers’ comments have been taken into account to improve the quality of our initial paper.

To facilitate the identification of the changes made within the manuscript, the modifications have been highlighted in yellow. Below are listed the changes in the manuscript, as well as our answers to the concerns raised by reviewers to facilitate your final decision.

We hope that such modifications might render our revised manuscript suitable for publication in Marine Drugs.

Yours sincerely, 

Mathilde Keck

Reviewer 2

The ARfD of a toxin is an estimate of the amount of substance in food that can be ingested in a period of 24 h or less without appreciable health risk to the consumer based on all known facts at the time of the evaluation.

If the goal of this study is the identification of potential points of departure to establish an acute oral reference dose (ARfD) why the experiment last 48h? Author should explain.

Answer: This study was performed according to the “Acute Toxic Class Method” from OECD guideline n° 423 (OECD 2001). This guideline mentioned “Animals are observed individually after dosing at least once during the first 30 minutes, periodically during the first 24 hours, with special attention given during the first 4 hours, and daily thereafter, for a total of 14 days. … However, the duration of observation should not be fixed rigidly. It should be determined by the toxic reactions, time of onset and length of recovery period”. We performed a 14-days pilot study that we now added to the manuscript in Appendix B (pages 15 to 17). This pilot study of acute oral administration of BTX-3 for 14 days in female Swiss mice showed that BTX-3 i) led to death in the first hour at 2,000 μg.kg-1, ii) induced a transient loss of weight after 24 h that was not observed after 48 h at 1,000 μg.kg-1, iii) caused a transient loss of temperature after 2 h for all tested doses that was not observed after 48 h, iv) resulted in a transient failure to grip test after 2 h at 750 and 1,000 μg.kg-1 that was not observed after 48 h, v) led to a transient decrease of water intake after 24 h at 1,000 μg.kg-1 that was not observed after 48 h, and vi) resulted in abnormal symptomatology during the first 4 hours at 1,000 μg.kg-1 that was not observed after 24 h. In conclusion, this pilot study indicated that abnormal parameters observed after BTX-3 administration were transient and all disappeared after 48 h. According to the results obtained from the pilot study, we designed the principal study to observed carefully the mice following the 4 h after administration, and then at 24 h, and 48 h after administration.

In addition,  the “Up-and-Down-Procedure” from OECD guideline n° 425 (OECD 2022) also indicated “Each animal should be observed carefully for up to 48 hours before making a decision on whether and how much to dose the next animal. That decision is based on the 48-hour survival pattern of all the animals up to that time”.

Therefore, guiding by the pilot study results and by the OECD guidelines, we designed the experiment for 48 h.

Also, most of the BTX3 effects were observed in the first 2 h but at 24 h decreased and at 48 were even not detected. The experimental time seems too long. In any case, it would be interesting to characterize clinical signs, biochemical or histopathological effects over the time course following acute exposure.

Answer: We would like to thank the reviewer for this relevant comment. Indeed, it would be interesting to characterize clinical signs, biochemical or histopathological effects over the time course following acute exposure. However, such characterization would lead to multiply the number of animals for each time to be investigated. We did not performed such characterization in order to be in compliance with 3Rs consisting to reduce the number of animals used. We choose to observe carefully the mice following the 4 h after administration, and then at 24 h, and 48 h after administration, before their euthanasia at 48 h (according the pilot study results and the OECD guidelines).

The study lacks control animals, which doesn’t allow to stablish the statistically significant toxin effect or to derive the appropriate points of departure for characterizing safe doses in humans. In this way, it is difficult to know if the lowest doses had an effect when compared to non-treated mice. For instance, mice body temperature not only varies throughout the day, but could also vary from group housing versus single housing animals. Therefore, it is suggested to perform an assay in control mice to assess whether the environmental conditions could be affecting some of the measured parameters.

Answer: The study design had been reviewed and approved by the experts from ANSES (as indicated in page 14, lines 490-492 of the manuscript). This is an original study where each animal was its own control since in Figures 1, 2, 3, and S1 we statistically compared each animal to its own value at T0. In Figures 1, 2, 3 and S1, we never compared the different dose with each other. Therefore, to perform an assay in control mice will have any impact of the results from Figures 1, 2, 3 and S1. The only comparisons between distinct doses were done for parameters observed after the euthanasia of mice at 48 h, i.e. for plasma chemistry analysis (Figure S2) and organs weight (Figures S3). As suggested by the reviewer, we now added to the supplemental figures S2 and S3 a group of control mice consisting to 4-weeks old male (n = 5) and female (n = 5) Swiss mice administered in the same condition as other mice with a solution of NaCl 0.9% with 5% DMSO. The addition of this group of control mice did not modify our results since no significant signs of perturbations of plasma chemistry were detected, with parameters remaining within standard thresholds (Figure S2). The only significant difference observed was between the glucose levels of male mice administered with 500 µg.kg-1 bw of BTX-3 versus female mice administered with 1,000 µg.kg-1 bw of BTX-3 that could probably be explained by randomization. Similarly, BTX-3, whatever the dose, did not induce any modification of organs weight (Figure S3). The only significant differences observed were between male versus female kidneys weight that could be explained by both sex-effect and randomization.

Besides, in some issues authors seems to be more focused in differences between male and female mice than in the BTX3 effect.

Answer: The principal objective of the study was to investigate the effect of a single oral administration of BTX-3 in mice. However, this is the first study of BTX-3 administered by acute oral gavage to both male and female mice, and we observed different sensitivity endpoints for males and females. We have taken into account the consideration of the reviewer by deleting the term related to differences between male and female in order to focus the results on BTX-3 effects (deletion of the term “In contrast” page 3, line 119; deletion of the term “However” page 4, line 155; deletion of the term “but no male” page 6, line 185; deletion of the term “but no female” page 6, line 195).

Also, it is unclear why authors selected 2h, 24 and 48 h for mice observation, must clarify the experimental reason for that.

Answer: Thanks to the pilot study, now added to the manuscript in Appendix B, we designed the principal study to observe carefully the mice following the 4 h after administration, and scored them at 2 h, 24 h, and 48 h after administration (see justification above).

To stablish the NOAEL or LOAEL of the critical end points it would be beneficial to monitor effects in different time points between 2h and 24 h.

Answer:  As mentioned below, thanks to the pilot study and according OECD guidelines, we designed the principal study to observe carefully the mice following the 4 h after administration, and scored them at 2 h, 24 h, and 48 h after administration (see justification above). To monitor effects in different time points between 2h and 24 h would lead to multiply the number of animals for each time to be investigated. We did not performed such characterization in order to be in compliance with 3Rs consisting to reduce the number of animals used.

Finally, there is no discussion, just a description of results. More interpretation and discussion of the findings are needed.

Answer: We thank the reviewer for this remark. We improve the discussion adding eight novel references (43 to 50). We described that a similar decrease in body temperature was also observed in several studies (new references 43 to 46) after intravenous or intraperitoneal or oral administration of BTX-2 or BTX-3 or ciguatoxins in mice or rat (page 9, lines 270-282). We also compared our study protocol to the recent study from Costas et al. (new reference 47) carried out in inbred female mice with BTX-3 diluted in ethanol after 15h of fasting duration and including not quantifiable parameters (pages 9-10, lines 291-307). We put the results from our study into context with the regulatory limits from other countries that currently regulate these toxins (page 10, lines 316-333).

Specific comments:

 Page 2. Line 73.

“emergent” should be changed for “emerging”.

Answer: The term “emergent” has been changed for “emerging” (page 2, line 68).

Page 2. Line 92-93.

Authors should explain why a single administration and 48 h observation period when ARfD is stablished in a period of 24 h

Answer: We have taken into account the reviewer comment by adding an Appendix B (pages 15 to 17) that described the pilot study, and we now justified the 48 h observation period of the principal study in the manuscript (page 12, lines 424-425).

Page 3. Line 103.

Since water intake is presented, was food intake also measured?

Besides, It could be interesting in the experiment to quantify urine and feces.

Answer: As point out by the reviewer, food intake, neither urine nor feces were quantified in this study. To measure precisely food intake, urine and feces, it would have been necessary to place mice into metabolic cages, leading to important stress and potential disruption of the symptomatology observed during scoring.

Page 3. Line 104-106.

It is stated that three male mice (1,500 μg.kg-1 bw) had lower water intake and they survived for 48 h, whereas in the following sentence it is stated that another three male mice had to be euthanized. In contrast, in Figure 4 it is indicated that this dose was administered to “n = 5 males”. This should be explained.

Answer: In the manuscript, it is stated that “All mice administered with different doses of BTX-3 showed a normal water intake”. Water intake was measured at 24h and 48h after administration. As it is stated in the manuscript, all surviving males showed a normal water intake at 24h and 48h after administration. However, three male out of the five administered had to be euthanized only 4 h after the administration because they showed critical end-points (important decrease of body temperature, full-body stiffening, uninterrupted convulsive movements of the jaw, and repeated tremors). Therefore, water intake was impossible to measure for these three male out of five administered. To avoid misunderstanding, we removed the first sentence related to water intake at the end of the paragraph (page 3, lines 109-110).

Page 3. Line 106-110.

Three male mice administered with 1,500 μg.kg-1 bw of BTX-3 had to be euthanized 4 h after the administration. Did the authors euthanize any male in the pilot study?

Answer: Pilot study was conducted only in female (not in male) mice to be in compliance with 3Rs rule. The administration of 2,000 μg.kg-1 bw of BTX-3 led to the rapid death (in the first hour) of the only female administered at this dose.

Page 3. Line 115-117.

How authors interpretate the transient low of weight after 24 h of male that received 1,000 μg.kg-1 bw of BTX-? May be 24 h is a crucial time that should study.

Answer: The question raised by the reviewer is interesting since the 24 h time of male that received 1,000 μg.kg-1 bw was the only time for male showing a loss of weight (the weight of male that received 1,500 μg.kg-1 bw could not be taken into account since 3 out of 5 male mice were euthanized at 4 h). We improved the results part of the manuscript by adding the following sentences in page 3, lines 131-134 “In addition, the administration of 1,000 µg.kg-1 bw led to a loss of weight in male mice 24 h after administration when compared to before administration. Overall, these results indicated that 24 h might be a crucial time for BTX-3 impact on animal weight”.

Page 3. Line 128-131.

Authors could not stablish that female mice seemed to be more sensitive to BTX-3 impact on loss of weight since the administration of the high dose of 1,500 μg.kg-1 bw of BTX-3 led to a significant loss of weight. Remember, that 3 of 5 male mice treated with 1,500 μg.kg-1 bw of BTX-3 were euthanized at 4 h.

 Answer: We would like to thank the reviewer for this relevant comment. We agree with reviewer 1 that, at the highest dose of 1,500 μg.kg-1 bw of BTX-3, we could not establish that female mice seemed to be more sensitive to BTX-3 impact on loss of weight since 3 out of 5 male mice treated with were euthanized at 4 h. Actually, we said that female mice seemed to be more sensitive to BTX-3 impact on loss of weight because all female mice treated with BTX-3, whatever the dose and even at the lowest dose of 100 μg.kg-1 bw, showed a decreased of body weight 24 h after administration. To avoid misunderstanding, we now clarify in the abstract (page 1, line 22) and the discussion (page 9, lines 266-267) of the manuscript that female mice seemed to be more sensitive to BTX-3 impact on the loss of body weight since the administration of the toxin, whatever the dose and even at the lowest dose of 100 µg.kg-1 bw, led to a significant loss of weight only in female mice 24 h after administration when compared to before administration.

Page 4. Line 142-144.

The significant differences in female body weight and body temperature decrease for 100 μg.kg-1 bw indicate that BTX-3 may still has an effect. Therefore, lower doses should be tested to assess at which dose the toxin does not cause adverse effects.

Answer: The significant differences in female body weight and body temperature come from the statistical analysis with default parameters. But these variation have no biological significance. In fact, the observed loss of body weight is less than 10%, a value that is commonly accepted in the literature as an adverse effect. This is the threshold we used to define the dose at which this 10% of reduction is observed. Similarly, the loss in body temperature comes from the statistical analysis with default parameters. But the variation has no biological significance. As stated in in Table 1, last column called “Comment”, we chose a threshold of 1°C as biologically significant. We search for a more scientific value in the literature, but found no information. For transparency of your choices and interpretation of our data, all our assumptions and hypothesis are explained in Table 1, last column called “Comment”. Therefore, despite these statistical differences, 100 μg.kg-1 bw of BTX-3 could be defined as the NOAEL. A novel publication dealing of “Acute toxicology report of the emerging marine biotoxin Brevetoxin 3 in mice” from Costas et al. was recently published in Food and Chemical Toxicology. This publication showed an oral LOAEL for BTX-3 of 100 μg.kg-1 bw and a NOAEL of 10 μg.kg-1 bw. However, in this study, inbred female mice were administered with BTX-3 diluted in ethanol after a 15 h of fasting. All these conditions could participate to increase the sensitivity of mice to BTX-3 regarding our experimental conditions (i.e outbred female mice administered with BTX-3 diluted in ≤ % DMSO after a 3 h of fasting). We now implemented the discussion of the manuscript (pages 9-10, lines 291-307) with the cited reference in order to discuss our results with regard to this recent study. 

Page 5. Line 170-172.

Since Authors found a significant difference between the glucose levels of male mice administered with 500 µg/kg and female mice administered with 1,000 µg/kg of BTX3 it would be beneficial to characterize food intake to contextualize the findings.

Answer:  We would like to thank the reviewer for this comment. It is important to note that, although a difference was observed between male mice administered with 500 μg.kg-1 and female mice administered with 1,000 μg.kg-1 of BTX-3, the glucose levels of these mice were not different from control group (now added to the supplemental Figure S2). Moreover, these levels were included between min and max thresholds and were relatively similar between the different groups of animals, suggesting that BTX-3 had no effect on glucose levels 48h after administration.

Page 6. Line 182-184.

Female administered with 500 µg/kg of BTX3 showed transient reduced muscle activity 24 h after administration, and this effect was not observed 48 h after administration.

It could be interesting to evaluate necropsy at 24h.

Answer: We would like to thank the reviewer for this relevant comment. As raised above, it would be interesting to characterize clinical signs, biochemical or histopathological effects at 24 h. However, such characterization would lead to increase the number of animals to use. We did not perform such characterization in order to be in compliance with 3Rs consisting to reduce the number of animals used. We choose to observe carefully the mice following the 4 h after administration, and then at 24 h, and 48 h after administration, before their euthanasia at 48 h (according to the pilot study results, and the OECD guidelines).

Page 6. 208-213.

Related to abnormal symptoms. Authors indicated that BTX3 impacts on symptomatology were rapid, appearing during the first 2 h. Did the authors observe the mice during the first 2h? As symptoms disappear 24 h after administration, did the authors find any abnormal symptoms between 2 and 24 h?

If 3 males were euthanized at 4h. What where the symptoms of the other mice at that time?

Answer: Mice were carefully observed following the 4 h after administration with a scoring performed 2 h after administration. Globally, the symptoms observed 2 h after administration were similar 4 h after administration. According to reviewer 1 comment, we clarified in the material and methods section in page 12, lines 428-429, and in Figure 4, and indicated that we carefully observed the mice following the 4 h after administration.

Page 8.

In Table 1 it is indicated 100 μg.kg-1 bw as the NOAEL for body weight and body temperature, whereas significant differences at 24 and 2 h respectively in Figures 1 and 2. Therefore, this dose should be not suggested as a NOAEL. This should be modified. Before defining 100 μg.kg-1 bw as NOAEL.

Answer: Regarding the reduction of body weight in females, the differences highlighted in Figure 1 come from the statistical analysis with default parameters, but the variation has no biological significance. It is commonly recognised that a reduction in body weight of 10% can be regarded as an adverse effect. This is the threshold we used to define the dose at which this 10% of reduction is observed. This is the case for the dose of 250 µg.kg-1, as stated in Table 1, last column called “Comment”. By definition, the dose below the LOAEL is a NOAEL.

The same explanation can be given for the reduction in body temperature in females. The differences highlighted in Figure 2 come from the statistical analysis with default parameters, but the variation has no biological significance. As stated in Table 1, last column called “Comment”, we choose a threshold of 1°C as biologically significant. We search for a more scientific value in the literature, but found no information.

For transparency of your choices and interpretation of our data, all our assumptions and hypothesis are explained in Table 1, last column called “Comment”.

Page 8. Line 232-233.

Authors should clarify the points of departure for stablishing ARfD. For instance, about muscle activity authors should explain why is 500 μg.kg-1 bw in males.

Answer: All the potential points of departure considered as candidates to be used to establish an ARfD are described in Table 1. For transparency of your choices and interpretation of our data, all our assumptions and hypothesis are explained in the last column called “Comment”. For muscle activity, based on Figure 3 Graph. b, at the dose of 100 and 250 µg.kg-1, the activity was normal. At the dose of 500 µg.kg-1, the activity is still normal for males, but not for females. By definition, the dose of 500 µg.kg-1 is a NOAEL for males, but not for females. For females, as an effect is observed at the dose of 500 µg.kg-1, this dose is a LOAEL and the NOAEL is the dose tested below, the dose of 250 µg.kg-1.

In the case of muscle activity, we presented the NOAEL for each sex instead of the NOAEL of the most sensitive sex as for other endpoints, because these data were also analysed for BMD modelling. We used the PROAST software with an option “sex effect”, as can be seen in the report provided as supplemental material. With the average BMD modelling, the lowest BMDL was obtained with the males (whereas the lowest NOAEL was for females). It is acknowledge that BMDLs are more robust values than NOAELs. For transparency, we presented both results (NOAELs and BMDLs) in our manuscript.

Page 9. Discussion.

There is no discussion, just a description of results with some contradictions. It is not clear if BTX3 is more toxic to male than female mice.

Answer:  We thank the reviewer for this remark. We now improved the discussion adding eight novel references (43 to 50) describing a similar decrease in body temperature (page 9, lines 270-282) and a similar acute toxicology of BTX-3 in mice (pages 9-10, lines 291-307). We also put our study into context with the regulatory limits from other countries that currently regulate BTX-3 (page 10, lines 316-333). As it is mentioned, in page 9, lines 262-263 “It is worth noting that BTX-3 administered at the highest dose of 1,500 µg.kg-1 bw was more toxic in males compared to females”.

Page 9. Line 284-286.

As authors pointed out that it would be relevant to identify the precise time of symptoms. Therefore, a study with more time points of test and symptoms evaluations must be performed.

Answer: Indeed, we pointed out that it would be relevant to identify the precise time of symptoms. However, it seems to us that this point is the subject of another study because such a study would need including a large number of animals and analyses.

Page 10. Materials and Methods. Line 299.

More details on toxin preparation and administration should be provided.

Answer: As suggested by the reviewer, we added more details on toxin preparation and administration in the material and methods section in pages 10-11, lines 353-376. In page 12, lines 422-424, it was indicated that mice administration was done by oral gavage with 10 µL.g-1 (199-312 µL per mouse) of BTX-3 with sterile plastic feeding tubes, 20 ga x 38 mm.

Page 10. Line 336-338.

It should be explained at which time point animals were caged individually.

Answer:  We have now clarified that animals were caged individually from T0 to T48h (page 12, line 414).

Page 11. Line 354-357.

Why the grip test was performed under only 5 sec after the grid was turned around? Please explain. More

Answer:  The grip test was performed under only 5 sec since the mice quickly started to explore their environment and they raised above the grid in less than 5 sec. Therefore, it was not useful to perform a longer grip test.

In the Section 4.4. Procedure for in vivo experiments, it is commented that a preliminary study was conducted to select the doses. It would be interesting that those results were also added to the manuscript.

Answer:  We have taken into account the reviewer comment by adding to the manuscript an Appendix B that described the pilot study (pages 15 to 17).

Supplementary Material.

Figure S3 significant differences in kidneys weight between treatments is shown. How could be this related to brevetoxins? It would be interesting to evaluate histological slices to determine whether these changes translate into organ damage. Moreover, organ weight of control animals would be helpful to address whether these changes are related to the toxin or are the result of sex dimorphism.

Answer:  As suggested by the reviewer, we now added a group of control mice consisting to male and female Swiss mice administered in the same condition as the other mice with a solution of NaCl 0.9% with 5% DMSO. The addition of this group of control mice did not modify our results on organs weight since BTX-3, whatever the dose, did not induce any modification of organs weight (Figure S3). The only significant differences observed were between the kidneys weight of male at 250 and/or 750 µg.kg-1 versus female at 250 and 500 µg.kg-1. These differences could be explained by both sex-effect and randomization since there was no statistical difference between the control group and the BTX-3 groups. Moreover, the publication from Costas et al. recently published in Food and Chemical Toxicology revealed no morphological alterations, no inflammation, nor necrotic areas, nor abnormal sizes or loss of normal structure was in any case. Regarding the reviewer comment, we now mentioned this point in the discussion of the manuscript (page 10, lines 305-306).

Reviewer 3 Report

Comments and Suggestions for Authors

This manuscript reports very important data on toxicity of brevetoxin-3, a marine biotoxin that is emerging in Europe, and for which limited data is available in terms of human intoxication or effects in experimental model animals. The manuscript is very well written and structured. The methodology is adequate, well detailed and accurate.

I have very minor issues otherwise I would recommend this manuscript for publication as is:

·       Line 253: replace “it was possible de calculate” with “it was possible to calculate”

·       Figure 4 is not mentioned in the text.

·       Line 288: should be moved to introduction

Author Response

We thank all the reviewers for their insightful assessment of our manuscript. All of the valuable reviewers’ comments have been taken into account to improve the quality of our initial paper.

To facilitate the identification of the changes made within the manuscript, the modifications have been highlighted in yellow. Below are listed the changes in the manuscript, as well as our answers to the concerns raised by reviewers to facilitate your final decision.

We hope that such modifications might render our revised manuscript suitable for publication in Marine Drugs.

Yours sincerely, 

Mathilde Keck

Reviewer 3

This manuscript reports very important data on toxicity of brevetoxin-3, a marine biotoxin that is emerging in Europe, and for which limited data is available in terms of human intoxication or effects in experimental model animals. The manuscript is very well written and structured. The methodology is adequate, well detailed and accurate.

I have very minor issues otherwise I would recommend this manuscript for publication as is:

  • Line 253: replace “it was possible de calculate” with “it was possible to calculate”

Answer: We thank the reviewer for this comment which we have taken into account by replacing “it was possible de calculate” by “it was possible to calculate” (page 9, line 256).

  • Figure 4 is not mentioned in the text.

Answer: We thank the reviewer for this remark, Figure 4 is now mentioned in page 12, line 424.

  • Line 288: should be moved to introduction

Answer: We thank the reviewer for this relevant recommendation which have taken into consideration by moving the cited sentence to introduction in page 2, lines 90-92.

Round 2

Reviewer 2 Report

Comments and Suggestions for Authors

The authors addressed most of the questions properly